# Identification and validation of intergeneric hybrids between *Saccharum officinarum* and *Erianthus rockii* using molecular and cytogenetic tools

**Gang Wang**[ORCID]**, Jianqiang Wang, Wei Zhang, Huanying Xu, Qinggan Liang, Yuanxia Qin, Qingdan Wu, Jiantao Wu, Cheng Fu, Feng Zhou, Yuxing An, Qinnan Wang**\*, **Hailong Chang**[ORCID]\*

Institute of Nanfan & Seed Industry, Guangdong Academy of Sciences, Guangzhou, Guangdong, China

\* hl2004@126.com (HC); wangqinnan66@163.com (QW)

## Abstract

Sugarcane (*Saccharum* spp.), a vital crop for sugar and bioenergy production, faces challenges in breeding due to its narrow genetic base and susceptibility to environmental stresses. To enhance genetic diversity, distant hybridization with wild relatives such as *Erianthus rockii*, known for its drought tolerance and strong ratooning ability, offers a promising strategy. However, identifying true intergeneric hybrids remains a critical challenge. This study identified true intergeneric hybrids from crosses between *S. officinarum* and *E. rockii* using tetra-primer ARMS-PCR and HRM curve analysis targeting SNPs in the nrDNA-ITS region, with genomic in situ hybridization (GISH) validating chromosome composition. The results confirmed 13 true hybrids among 16 progeny, showing a chromosome inheritance pattern of 40 chromosomes derived from *S. officinarum* and 15 from *E. rockii*. Hybrid plants, while exhibiting traits from both parents, displayed inferior yield and sugar content compared to the maternal parent, indicating the necessity for further backcrossing to improve agronomic performance. This study pioneers the application of ARMS-PCR and HRM in identifying *E. rockii* hybrids and underscores their potential in advancing sugarcane breeding by facilitating the introgression of beneficial traits from wild relatives.

## Introduction

Sugarcane (*Saccharum* spp.) is a globally significant crop valued for its sugar and bioenergy potential, classified under the Poaceae family and Andropogoneae tribe. Cultivated primarily in tropical and subtropical regions across over 100 countries [1]. Sugarcane cultivars are highly complex polyaneuploid interspecific hybrids derived from *S. officinarum* ($2n = 8x = 80$, $x = 10$), known for high sucrose and low fiber content, and *S. spontaneum* ($2n = 4x\text{-}16x = 40\text{–}128$, $x = 8$, 9, or 10), recognized for stress resistance and high fiber [2,3]. However, breeding programs face significant

**Data availability statement:** All relevant data are within the manuscript and its Supporting Information files.

**Funding:** This research was funded by the GDAS' Project of Science and Technology Development (2022GDASZH-2022010102) and the GDAS's Project of Technical Innovation and Incubation Service Platform Construction (2021GDASYL-20210301001). The funders had no role in study design, data collection and analysis, decision to publish, or preparation of the manuscript.

**Competing interests:** The authors have declared that no competing interests exist.

challenges due to the limited genetic diversity arising from reliance on a narrow germplasm base, despite more than a century of sugarcane breeding [4]. Maintaining and utilizing sugarcane germplasm diversity is vital to enhance sugar yield and improve stress tolerance. Introgressing genes from wild relatives via distant hybridization is a critical strategy for broadening the genetic base and increasing adaptability [5]. The 'Saccharum complex', which encompasses related genera such as *Erianthus*, *Narenga*, *Miscanthus*, and *Sclerostachya*, serves as an invaluable genetic resource [6]. Among these, *Erianthus rockii* stands out with key traits such as drought and cold tolerance, brown rust resistance, and good ratooning ability, making it integral to sugarcane breeding programs [7,8].

Crosses between *S. officinarum* with *E. rockii* have yielding 16 progeny in recent years. Identifying true hybrids in intergeneric crosses is crucial for sugarcane breeding but poses challenges because phenotypic observations are often unreliable, being heavily influenced by environmental factors. Molecular markers offer a reliable alternative, with RAPD, AFLP, ISSR, and SSR previously applied to identify *Saccharum* species and related genera [6,7,9,10]. However, these makers often generate complex amplification patterns, complicating analysis. In contrast, single nucleotide polymorphisms (SNPs), known for their high precision, are increasingly utilized in crop genetics and marker-assisted breeding [11]. Among over ten SNP genotyping methods, tetra-primer amplification refractory mutation system-polymerase chain reaction (ARMS-PCR) stands out as a simple yet cost-effective method for SNP-based genotyping [12]. It employs allele-specific primers to amplify specific SNP alleles, enabling straightforward genotype identification [13]. This technology has been extensively utilized for the analysis and identification of germplasm genotypes in crops, including sweet potato and various medicinal plants [14–16]. High-resolution melting (HRM) curve analysis, another SNP-based method, analyzes genetic variation by identifying distinct melting temperature profiles [17]. With advantages such as super-sensitivity, repeatability, closed-tubed operation, and cost-effectiveness, HRM finds extensive applications in genotyping plant species, encompassing mutation detection and marker-assisted selection [17–19]. However, these two SNP genotyping methods have not yet been applied to the identification of *E. rockii* hybrids. Furthermore, examining chromosome transmission behavior in intergeneric hybrids is essential for the effective application of introgression programs. The genomic in situ hybridization (GISH) technique serves as a tool to investigate chromosomal structure, exchange, and inheritance dynamics across generations [20]. As a widely used molecular cytogenetic technology, GISH facilitates the identification of chromosome composition in interspecific and intergeneric hybrids from diverse species, offers valuable insights for sugarcane breeding strategies [21,22].

The internal transcribed spacer (ITS) region of the 18S-5.8S-26S nuclear ribosomal DNA (nrDNA) evolves rapidly and is widely used for DNA barcoding and phylogenetic analyses in plants [23,24]. It can be readily amplified and sequenced using universal PCR primers. ITS sequences, because of their substantial variation, have been utilized to develop molecular markers for identifying medicinal herb germplasms [25] and distinguishing orchid interspecific hybrids [26] as well as intergeneric hybrids

between *Argyranthemum frutescens* and *Rhodanthemum gayanum* [27]. Moreover, tetra-primer ARMS-PCR and HRM, based on stable SNPs in the nrDNA-ITS region, have been successfully utilized to identify *Saccharum* species and certain medicinal herb germplasms [28–32]. However, these applications for identifying *E. rockii* hybrids remain underexplored. In this study, we aimed to identify true hybrids from intergeneric crosses between *S. officinarum* and *E. rockii* using molecular authentication. Tetra-primer ARMS-PCR and HRM curve assays, targeting SNP allelic variations in the nrDNA-ITS region, were utilized to identify intergeneric hybrids, with subsequent confirmation through GISH analysis. These identified hybrids serve as invaluable genetic resources for sugarcane breeding, facilitating the incorporation of *E. rockii*'s beneficial traits into cultivated sugarcane.

## Materials and methods

### Plant materials and growth conditions

Two species, *S. officinarum* 'Badila' and *E. rockii* 'Yunnan 15-7', were used in the present study. The cross involved 'Badila' as the female parent and 'Yunnan 15-7' as the male parent. In this study, when the pollen shedding of 'Badila' was about to begin, the tassels were treated with hot water at 50 °C for 5 minutes for emasculation and nurtured in greenhouses for 2 days. Fresh pollen from 'Yunnan 15-7' was collected at 8:30 AM daily and manually applied to the emasculated tassels of 'Badila' over a 5-day period. The hybrid tassels were maintained in the greenhouse until hybrid seeds were produced. Sixteen $F_1$ hybrids were obtained from an intergeneric cross between 'Badila' and 'Yunnan 15-7'. All the $F_1$ plants and their parents were grown at the Hainan Sugarcane Breeding Station of the Institute of Nanfan and Seed Industry of Guangdong Academy of Sciences under natural growth conditions. Additionally, nine nrDNA-ITS sequences from *S. officinarum* and *E. rockii* were downloaded from the NCBI nucleotide archive.

### DNA extraction and PCR amplification of nrDNA-ITS region

Plant leaves were frozen in liquid nitrogen, ground into fine powder, and their genomic DNA was extracted using the traditional CTAB method. The quality and concentration of the DNA were evaluated using 0.8% agarose gel and a Thermo Nanodrop 2000 spectrophotometer. DNA samples were then diluted to a concentration of 20 ng/µl with deionized water for PCR amplification. The nrDNA-ITS region of the two parents, which includes the ITS1, 5.8S, and ITS2 regions, was amplified using the universal primers ITS1 (5′-TCCGTAGGTGAACCTGCGG-3′) and ITS4 (5′-TCCTCCGCTTATTGATATGC-3′), located at the 3′ end of 18S rDNA and the 5′ end of 28S rDNA, respectively. PCR reaction mixtures (20 µl) were prepared on ice, containing 20 ng of template DNA, 0.5 µM of each primer, and 10 µl of 2 × Super Pfx MasterMix (CWBIO, Taizhou, China), and amplified using a Bio-Rad C1000 Touch™ Thermal Cycler (CFX96™ Optics Module, Bio-Rad, Hercules, CA, USA). The amplification profile consisted of a pre-denaturation cycle at 98 °C for 3 minutes, followed by 35 cycles of 98 °C for 10 seconds, 55 °C for 15 seconds, 72 °C for 15 seconds, and a final extension at 72 °C for 10 minutes. The PCR products were tested by 1.0% agarose gel electrophoresis and purified using the Omega EZNA gel extraction kit (Omega, USA). The purified products, following the addition of adenine (A), were cloned into the pMD19-T-vector (Takara, Dalian, China) and transformed into *E. coli* DH5α competent cells (Weidibio, Shanghai, China). Recombinant clones were grown in LB medium supplemented with ampicillin (100 µg/ml). Five clones per sample were selected for bidirectional sequencing by Tsingke Biotechnology Co., Ltd. (Beijing, China).

### ITS sequences analysis and alignment

The nrDNA-ITS sequences were analyzed and compared using the NCBI BLAST tool (http://www.ncbi.nlm.nih.gov/BLAST/). The locations of ITS1, 5.8S, and ITS2 regions were identified through comparison with reference sequences from the NCBI database. The nrDNA-ITS sequences of the two parental species were aligned using ClustalW 2.0 and manually adjusted for accuracy. The length (bp) and variable site information of ITS1, 5.8S rDNA, and ITS2 sequences were analyzed using DNAMAN 6.0 software.

## ARMS-PCR strategy

Tetra-primer ARMS-PCR primers were designed based on the nrDNA-ITS sequences of *S. officinarum* 'Badila' and *E. rockii* 'Yunnan 15-7', following the method described by Medrano and de Oliveira [12] for identifying intergeneric hybrids. Hybrids identification was carried out using tetra-primer ARMS-PCR. PCR reaction mixtures (20 μl) were prepared on ice, containing 20 ng of template DNA, 0.20 μM of each outer primer (ITS1 and RO), 0.20 μM of each inner primer (FI-Er and RI-So), and 10 μl of 2× Taq Master Mix (Vazyme, Nanjing, China). Reactions were performed using the Bio-Rad C1000 Touch™ Thermal Cycler. The PCR reaction conditions were as follows: (1) 1 pre-denaturation cycle at 94 °C for 5 minutes, (2) 5 cycles of denaturation at 94 °C for 30 seconds, annealing at 62 °C for 30 seconds (decreasing by 1 °C per cycle) and extension at 72 °C for 45 seconds, (3) 30 cycles of denaturation at 94 °C for 30 seconds, annealing at 57 °C for 30 seconds and extension at 72 °C for 45 seconds, (4) a final extension at 72 °C for 10 minutes. The PCR products were electrophoresed on a 1.0% agarose gel with SuperRed staining (Biosharp, Beijing, China). The image was captured using the Bio-Rad ChemiDoc XRS+ system and analyzed with the Image Lab™ software (Version 5.2.1, Bio-Rad, Hercules, CA, USA).

## High-resolution melting (HRM) curve analysis

To conduct the HRM assay, a primer set (HRM-SEF and HRM-SER) was designed in a conserved region based on nrDNA-ITS sequence alignment using Oligo 7 software, following the method previously described by Grazina et al. [17]. Conventional PCR was performed prior to HRM to ensure the amplification of a single band. PCR reaction mixtures (20 μl) were prepared on ice, containing 20 ng of template DNA, 0.5 μM of each primer (HRM-SEF and HRM-SER), and 10 μl of 2× Taq Master Mix (Vazyme, Nanjing, China), and amplified using a PCR instrument. The amplification profile consisted of 1 pre-denaturation cycle at 94 °C for 5 minutes, followed by 35 cycles of 30 seconds at 94 °C, 30 seconds at 60 °C, 20 seconds at 72 °C, and a final extension at 72 °C for 5 minutes. The PCR products were tested by 1.0% agarose gel electrophoresis.

Upon completion of the primer optimization using conventional PCR, HRM analysis was conducted using the Light-Cycler® 480 II System (Roche, Basel, Switzerland). Briefly, the reaction mixture contained 20 ng of template DNA, 0.25 μM of each primer (HRM-SEF and HRM-SER), 0.2 mM of dNTPs, 2 ul of 10× TransTaq® HiFi Buffer I, 1 unit of TransTaq® HiFi DNA Polymerase, and 1 ul of 20× LyGreen (Life-iLab, Shanghai, China), making a total volume of 20 μl. The reaction conditions included: (1) 1 pre-denaturation cycle at 95 °C for 10 minutes; (2) 40 cycles of denaturation at 95 °C for 10 seconds, annealing at 60 °C for 20 seconds, and extension at 72 °C for 20 seconds; and (3) HRM analysis with denaturation at 95 °C for 1 minutes, annealing at 40 °C for 20 seconds, and a melting curve analysis from 65 °C to 95 °C (recording 25 points/°C). Afterwards, the genotypes were determined by examining the normalized melt plots using the LightCycler® 480 Melt Genotyping Software module (Version 1.5, Roche, Basel, Switzerland).

## Genomic in situ hybridization procedure

Genomic DNA from *E. rockii* 'Yunnan 15-7' was labeled with FITC-12-dUTP (green) and genomic DNA from *S. officinarum* 'Badila' was labeled with Cy3-dUTP (red) using the Nick Translation Kit (Roche, Basel, Switzerland). Chromosome preparation, spreading, and GISH experiments were performed following the protocols described by D'hont et al. [33] and Li et al. [29], with minor modifications. A hybridization solution containing the two gDNA probes was prepared and dropped onto the dried slide. Hybridization was conducted overnight in a humidified incubation chamber at 37 °C. After hybridization, coverslips were gently removed, and the slides were washed three times for 5 minutes in 1× PBS. The dried slides were counterstained with DAPI, and well-spread metaphase images were captured using an AxioScope A1 Imager fluorescent microscope with an AxioCam MRc5 camera (Carl Zeiss, Göttingen, Germany). Images were analyzed using the AxioVision v.4.7 imaging software (Carl Zeiss, Göttingen, Germany).

## Results

### PCR amplification of nrDNA-ITS

Numerous *S. officinarum* and *E. rockii* nrDNA-ITS sequences have been published and deposited in GenBank. Additionally, nine nrDNA-ITS sequences from *S. officinarum* and *E. rockii* were downloaded from the NCBI nucleotide archive. As shown in Fig 1, the ITS1-5.8S-ITS2 rDNA region of *S. officinarum* had ITS1 sequences ranging from 206 to 207 bp, 5.8S region of 164 bp, and ITS2 sequences ranging from 217 to 219 bp. In contrast, the ITS1-5.8S-ITS2 rDNA region of *E. rockii* had fixed lengths of 207 bp, 164 bp, and 218 bp for ITS1, 5.8S, and ITS2 regions, respectively. Notably, the ITS1-5.8S-ITS2 rDNA region of *E. rockii* exhibited greater sequence length conservation compared to *S. officinarum*. The nrDNA-ITS regions (including ITS1-5.8S-ITS2) of *S. officinarum* 'Badila' and *E. rockii* 'Yunnan 15-7' were amplified using the ITS1 and ITS4 primer set, producing a single, intense band of approximately 677 bp in both species (Fig 2).

### Sequence analysis of ITS1-5.8S-ITS2

The ITS1-5.8S-ITS2 region of *S. officinarum* 'Badila' and *E. rockii* 'Yunnan 15-7' was found to be 589 bp in length, with ITS1, 5.8S, and ITS2 regions measuring 207 bp, 164 bp, and 218 bp, respectively (Fig 3). Alignment of all 589 sites in the

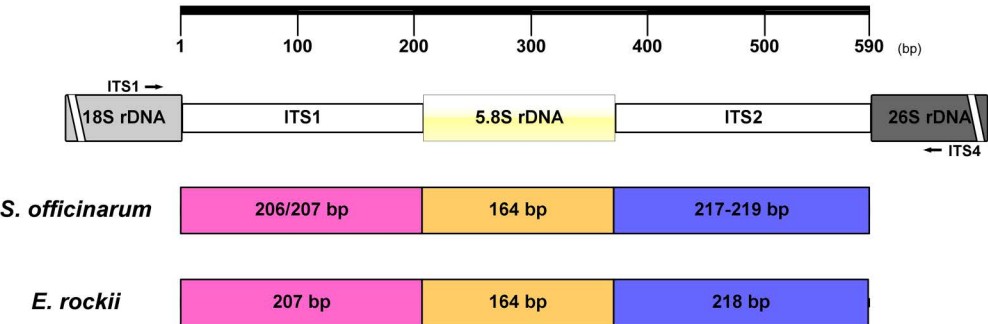

**Fig 1. Schematic representation of the nrDNA-ITS region from *S. officinarum* and *E. rockii*.** Numbers inside the bars represent the lengths of the ITS1, 5.8S rDNA, and ITS2 regions of *S. officinarum* and *E. rockii* ITS sequences retrieved from the NCBI database. The black arrow indicates the position of the universal primers used for amplifying the ITS region.

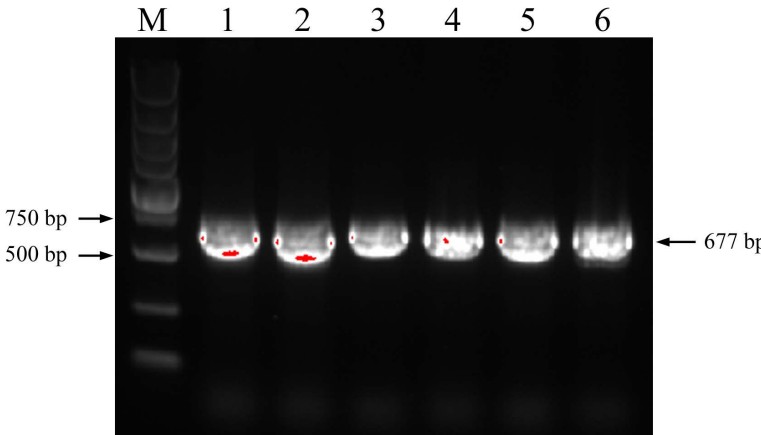

**Fig 2. Electrophoretogram of nrDNA-ITS PCR products.** Lane M: DL5000 DNA marker; lanes 1-3: *S. officinarum* 'Badila'; lanes 4-6: *E. rockii* 'Yunnan 15-7'.

sequences of the two species identified 556 conserved sites and 33 variable sites (Fig 3). The distribution of the 33 variable sites in the ITS region revealed that ITS1 had the highest variability (19 sites), followed by ITS2 (11 sites) and 5.8S rDNA (3 sites). The 5.8S region was the most conserved, with 161 out of 164 sites (98.17%) conserved, followed by ITS2 (207 out of 218, 94.95%) and ITS1 (188 out of 207, 90.82%). As shown in Fig 3, haplotype diversity analysis revealed four haplotypes among the two parental species, with three haplotypes identified in *S. officinarum* 'Badila' and one in *E. rockii* 'Yunnan 15-7'.

## Tetra-primer ARMS-PCR identification of true and false F$_1$ hybrids

Based on interspecific differences and conserved intraspecific SNP sites, species-specific SNP alleles for *S. officinarum* and *E. rockii* were identified at position 370 and 371 bp in the nrDNA-ITS region (Fig 3). These SNPs were used to design

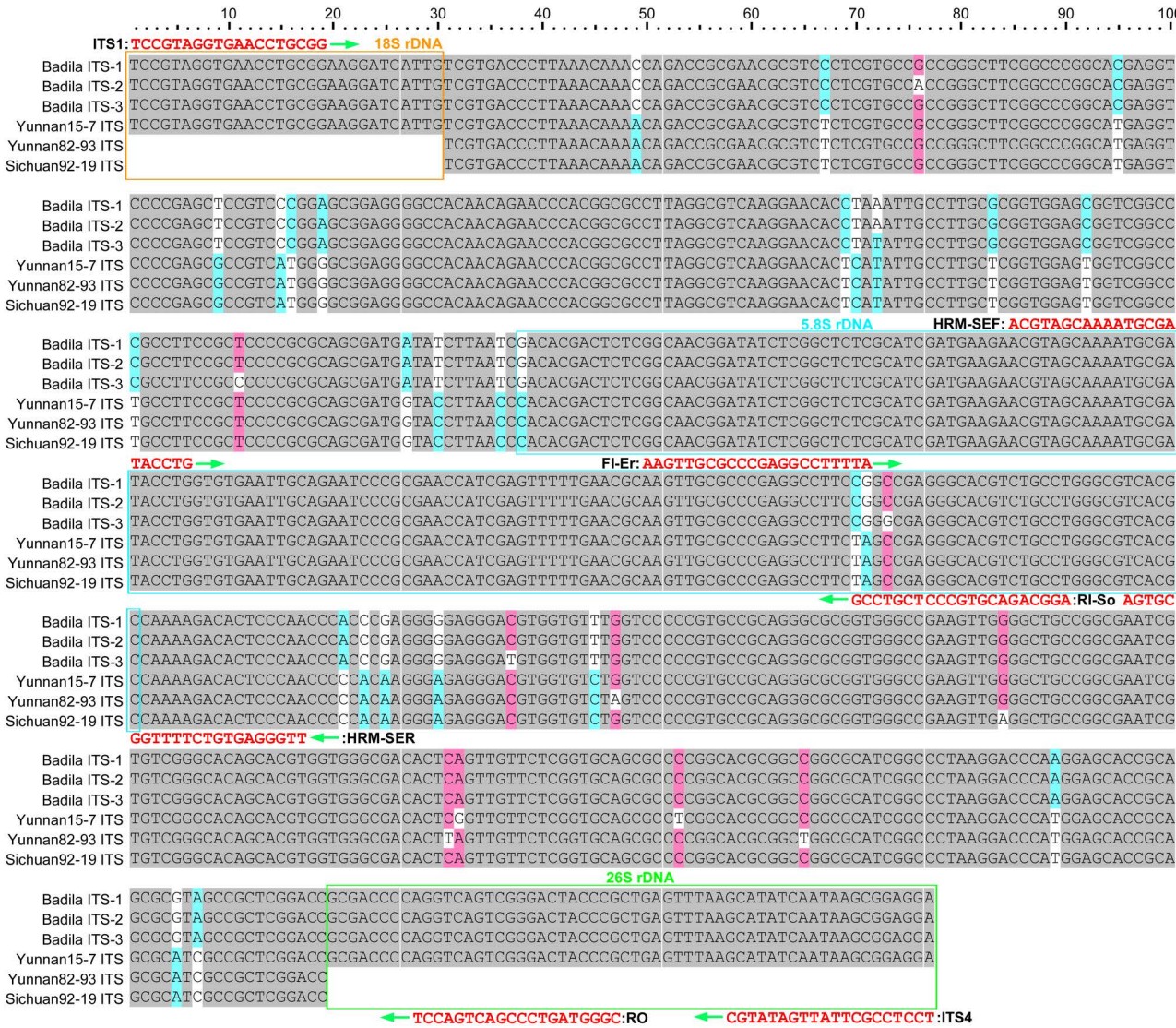

**Fig 3. Multiple sequence alignment of the nrDNA-ITS region from four clones of *S. officinarum* and *E. rockii*, including the positions and directions of primers used in this study.** Gray boxes show the same sequences of the two species. Other color boxes indicate the SNP sites. ITS sequences of 'Yunnan 82-93' and 'Sichuan 92-19' clones were downloaded from NCBI.

two species-specific primers using Oligo 7. The forward primer FI-Er was designed specifically to amplify *E. rockii* and did not amplify *S. officinarum*. Furthermore, the third-to-last base at the 3' end of the primer was deliberately designed to introduce a mismatch with any sequence (Fig 3). The reverse primer RI-So was specifically designed for *S. officinarum*, with the fourth-to-last base of its 3' end deliberately mismatched with any sequence (Fig 3). In addition, as shown in Table 1, FI-Er contained a TA-CG substitution at the last two base of the 3' primer end in *E. rockii* relative to *S. officinarum*, whereas RI-So had a CG-TA substitution at the last two base of the 3' primer end in *S. officinarum* relative to *E. rockii*. As shown in Fig 4a, these species-specific primers, incorporating mismatches at their 3' end, facilitated the preferential amplification of one allele over another and were designated inner primers. Two universal primers, ITS1 and RO, were shared by both *S. officinarum* and *E. rockii* and were designated as outer primers in the tetra-primer ARMS-PCR technique. As shown in Fig 4b, a 390 bp PCR product was expected to be amplified from *S. officinarum* using the Forward Outer and Reverse Inner primers. In contrast, a 298 bp PCR amplicon was expected to be amplified from *E. rockii* using the Forward Inner and Reverse Outer primers. Additionally, a 647 bp PCR product is also expected from both *S. officinarum* and *E. rockii* using Forward Outer and Reverse Outer primers as an internal control. The three PCR products differed significantly in length, enabling their clear differentiation using agarose gel electrophoresis.

The tetra-primer ARMS-PCR was used to amplify the DNA from two parents and all suspected $F_1$ hybrids. As shown in Fig 5, both *S. officinarum* and *E. rockii* parents exhibited a 647 bp PCR product, while the *S. officinarum* parent displayed a unique 390 bp product of and the *E. rockii* parent a 298 bp product. Thirteen $F_1$ progeny from the *S. officinarum* × *E. rockii* ('Badila' × 'Yunnan 15-7') cross exhibited both PCR products specific to *S. officinarum* and *E. rockii*, confirming their status as true $F_1$ hybrids from this intergeneric cross.

## HRM assay identification of true and false $F_1$ hybrids

Based on the alignment of the nrDNA-ITS sequences (Fig 3), a primer pair (HRM-SEF: 5'-ACGTAGCAAAATGCGATACCTG-3' and HRM-SER: 5'-TTGGGAGTGTCTTTTGGCGTGA-3') was selected for HRM analysis. The expected PCR products contained two adjacent SNPs (TA/CG) specific to the respective species, enabling their discrimination. As shown in Fig 6a, using the HRM-SEF and HRM-SER primer set, a single, intense 133 bp ITS fragment was amplified from both parents and the $F_1$ intergeneric hybrids. For HRM analysis, the melting features of partial ITS amplicons were assessed by plotting three different curves for the two parents and the $F_1$ intergeneric hybrids in the normalized melting curves (Fig 6b) and the normalized and temperature-shifted difference curves (Fig 6c). These curves produced unique and easily distinguishable plots for the two parents and the $F_1$ intergeneric hybrids. As shown in Fig 6b&c, melting curve patterns corresponded precisely with genotype classifications: samples from *E. rockii* displayed a green pattern, $F_1$ intergeneric hybrids exhibited a blue pattern, and *S. officinarum* and false hybrids showed a red pattern. These results demonstrated that nrDNA-ITS barcoding combined with HRM analysis could effectively distinguish the genotypes of parental homozygotes, heterozygotes of $F_1$ intergeneric hybrids, and false hybrids.

**Table 1. Sequences of the two species-specific primers developed in this study compared with the reported ITS sequences of *S. officinarum* and *E. rockii*.**

| Species name | Haplotype of ITS sequence | FI-Er (5'-3')  AAGTTGCGCCCGAGGCCTTTTA | RI-So (5'-3')  AGGCAGACGTGCCCTCGTCCG | GenBank accession No. | | | |
|---|---|---|---|---|---|---|---|
| *S. officinarum* | SoITS | AAGTTGCGCCCGAGGCCTTC**CG** | AGGCAGACGTGCCCTCGGC**CG** | AB250691 | AB250692 | AY116284 | KT907428 |
| | | | | KT907430 MN519273 SRR528718 | | | |
| *E. rockii* | ErITS | AAGTTGCGCCCGAGGCCTTC**TA** | AGGCAGACGTGCCCTCGGC**TA** | AF345216 AF345217 | | | |

Bold letter indicate the SNP used to design species-specific primers in this study.

## Chromosome composition of the F₁ intergeneric hybrids by GISH

To further investigate the genomic composition of the F₁ intergeneric hybrids, Thirteen F₁ hybrids were selected based on their confirmation as true hybrids via tetra-primer ARMS-PCR and HRM identification. The genomic compositions of these

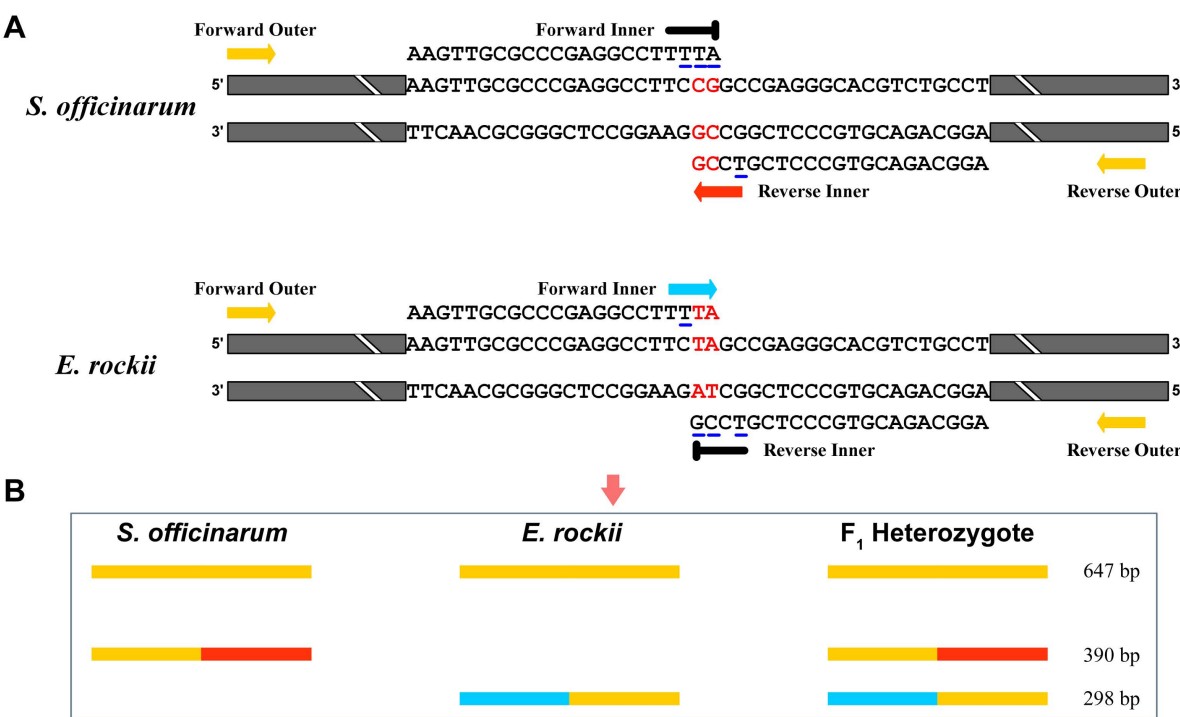

**Fig 4. Schematic representation of the tetra-primer ARMS-PCR assay for SNP genotyping (A) and a diagram illustrating the expected PCR product sizes for *S. officinarum*, *E. rockii*, and their F₁ hybrids on the agarose gel (B). (A)** Targeted SNPs are highlighted in red, and the underlined bases in the Forward Inner primer and the Reverse Inner primer represent mismatches with the primer binding sequences. Colored thick arrows indicate the amplification direction of primers, and black thick T-symbols denote ineffective primer amplification. **(B)** Different colors indicate distinct primers involved in the PCR reaction.

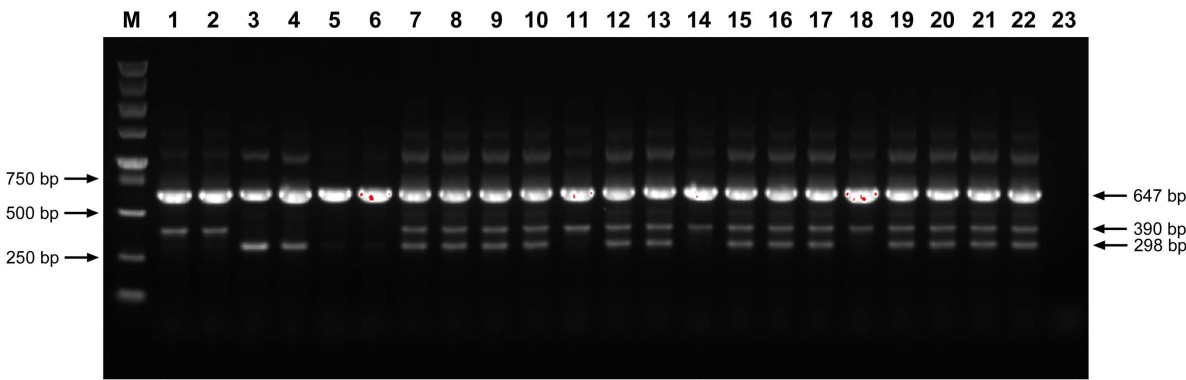

**Fig 5. Electrophoretogram of tetra-primer ARMS-PCR products for identifying intergeneric hybrids from *S. officinarum* × *E. rockii*.** Lane M: DL5000 DNA marker; lanes 1-2: *S. officinarum* 'Badila'; lanes 3-4: *E. rockii* 'Yunnan 15-7'; lanes 5-6: *N. porphyrocoma* 'Guangdong 32'; lanes 7-22: sixteen suspected F₁ intergeneric hybrids from 'Badila' × 'Yunnan 15-7'; lane 23: ddH₂O.

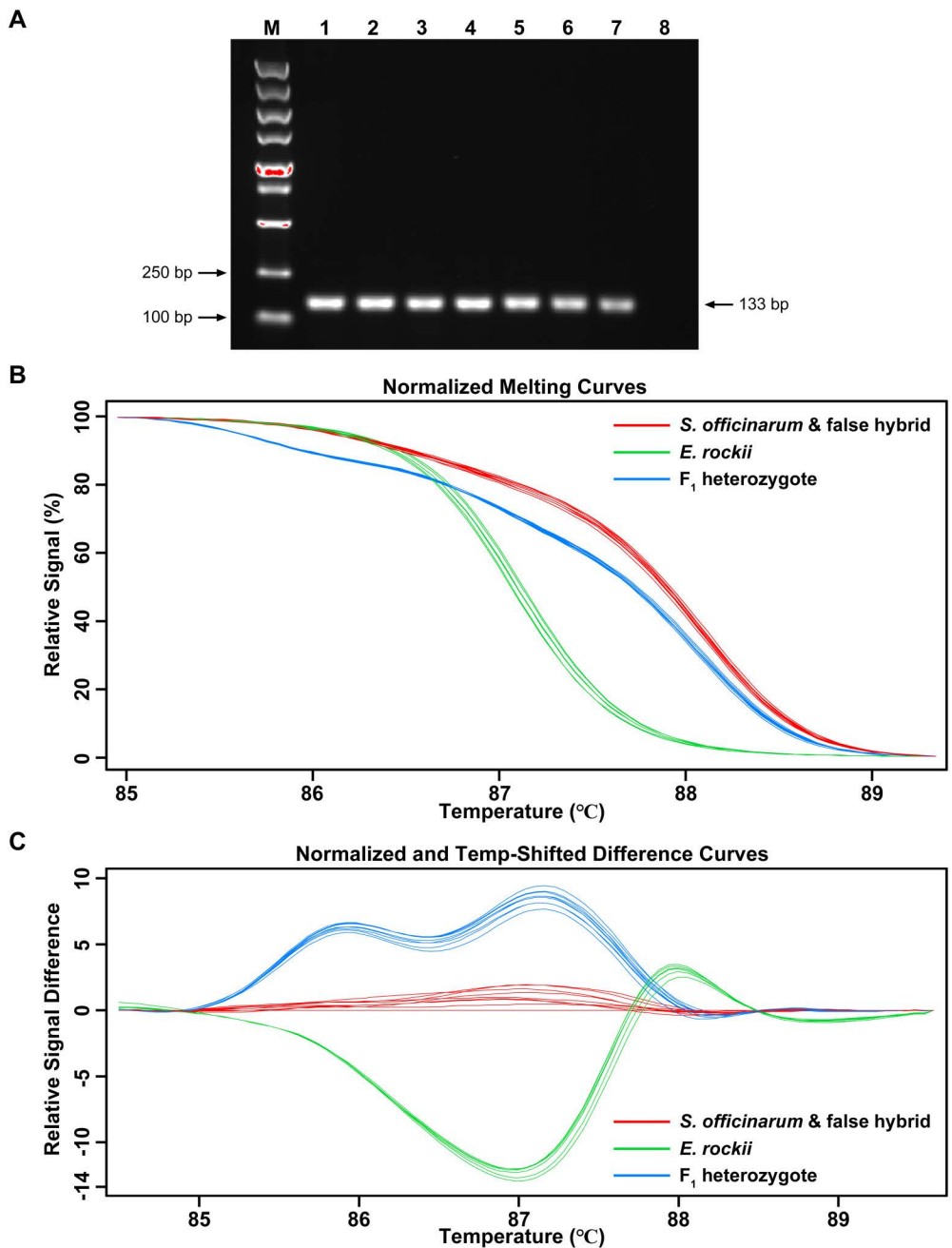

**Fig 6. HRM curve analysis using the designed ITS primers to analyze *S. officinarum*, *E. rockii*, and their F$_1$ hybrids. (A)** PCR results using primers HRM-SEF and HRM-SER. Lane M: DL5000 DNA marker; lanes 1-2: *S. officinarum* 'Badila'; lanes 3-4: *E. rockii* 'Yunnan 15-7'; lanes 5-7: suspected F$_1$ intergeneric hybrids from 'Badila' × 'Yunnan 15-7'; lane 8: ddH$_2$O. **(B)** HRM normalized melting curves and **(C)** normalized and temperature-shifted difference curves using *S. officinarum* 'Badila' as a baseline for the ITS amplicons using primers HRM-SEF and HRM-SER. The HRM software groups samples based on melting curves reflecting the nucleotide composition of the amplified PCR fragments. Similar melting curves are displayed in the same color: red represents *S. officinarum* and false hybrids; green represents *E. rockii*; blue represents genuine hybrids.

hybrids were determined using GISH. Previous studies have reported that *S. officinarum* has 2n = 80 chromosomes, while *E. rockii* has 2n = 30 chromosomes. As shown in Fig 7, GISH analysis revealed that $F_1$ hybrids from *S. officinarum* × *E. rockii* ('Badila' × 'Yunnan 15-7') contained 15 *E. rockii* chromosomes and 40 *S. officinarum* chromosomes in mitotic metaphase. These results demonstrated that these hybrids' genome contain both *S. officinarum* and *E. rockii*, and the chromosomal transfer model in the $F_1$ generation of *S. officinarum* × *E. rockii* followed an n + n pattern.

## Main agronomic traits of $F_1$ hybrids

As shown in Fig 8, the $F_1$ intergeneric hybrid plants exhibited distinct paternal characteristics in their gross morphology. The stalk diameter of the hybrids was intermediate between the female parent and the male parent, while the leaf morphology and stalk color resembled those of the female parent 'Badila'. The characteristics of the $F_1$ intergeneric hybrid plants are summarized in Table 2. The average phenotypic characteristics of the $F_1$ hybrids were generally intermediate between the parental values. The stalk length of the hybrids (163.87 cm) was approximately average of the parental values. The hybrids had an internode count of 18.47, compared to 20.17 in *S. officinarum* 'Badila' and 12.92 in *E. rockii* 'Yunnan 15-7'. The hybrids had a leaf length of 108.40 cm and a stalk diameter of 24.73 mm, compared to 136.27 cm and 39.60 mm in 'Badila', and 47.80 cm and 6.01 mm in 'Yunnan 15-7'. However, the leaf width of the hybrids (5.29 cm) was almost the same as that of the female parent 'Badila' (5.47 cm). In addition, the juice brix value of the hybrids (13.06%) was about 65% that of the female parent 'Badila' (20.79%).

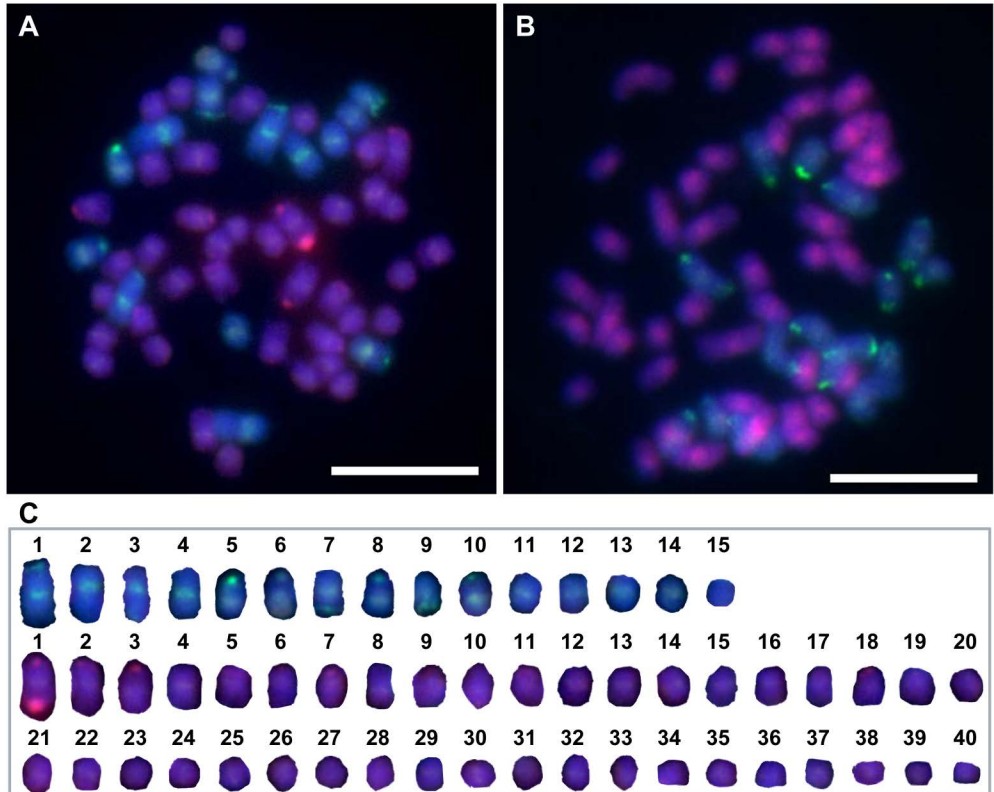

**Fig 7. GISH analysis of $F_1$ intergeneric hybrids between *S. officinarum* and *E. rockii*. (A)** C19-19; **(B)** C19-24; **(C)** Statistical results for chromosome counts of *S. officinarum* and *E. rockii* based on analysis from **(A)**. *S. officinarum* chromosomes are shown in red and *E. rockii* chromosomes in green. Bars = 10 μm.

## Discussion

Accurately identifying intergeneric hybrids remains a key challenge in sugarcane germplasm innovation. In this study, we identified stable mutations at base positions 370 and 371 within the nrDNA-ITS sequence of *S. officinarum* and *E. rockii* and designed species-specific primers for tetra-primer ARMS-PCR based on these SNPs. The optimized primers (ITS1, RO, FI-Er, and RI-So) successfully identified intergeneric hybrids. Additionally, ITS primers were developed to conduct HRM curve analysis, further validating the hybrids as genuine. This study represents the first application of tetra-primer ARMS-PCR and HRM for identifying hybrids of *S. officinarum* and *E. rockii*. GISH analysis corroborated the molecular findings, offering reliable methods for hybrid verification and advancing sugarcane germplasm innovation.

Distant hybridization remains a foundational approach to sugarcane germplasm innovation and is widely employed in crop improvement. Intergeneric hybrids of *Saccharum* with *E. rockii* have been previously reported for enhancing genetic

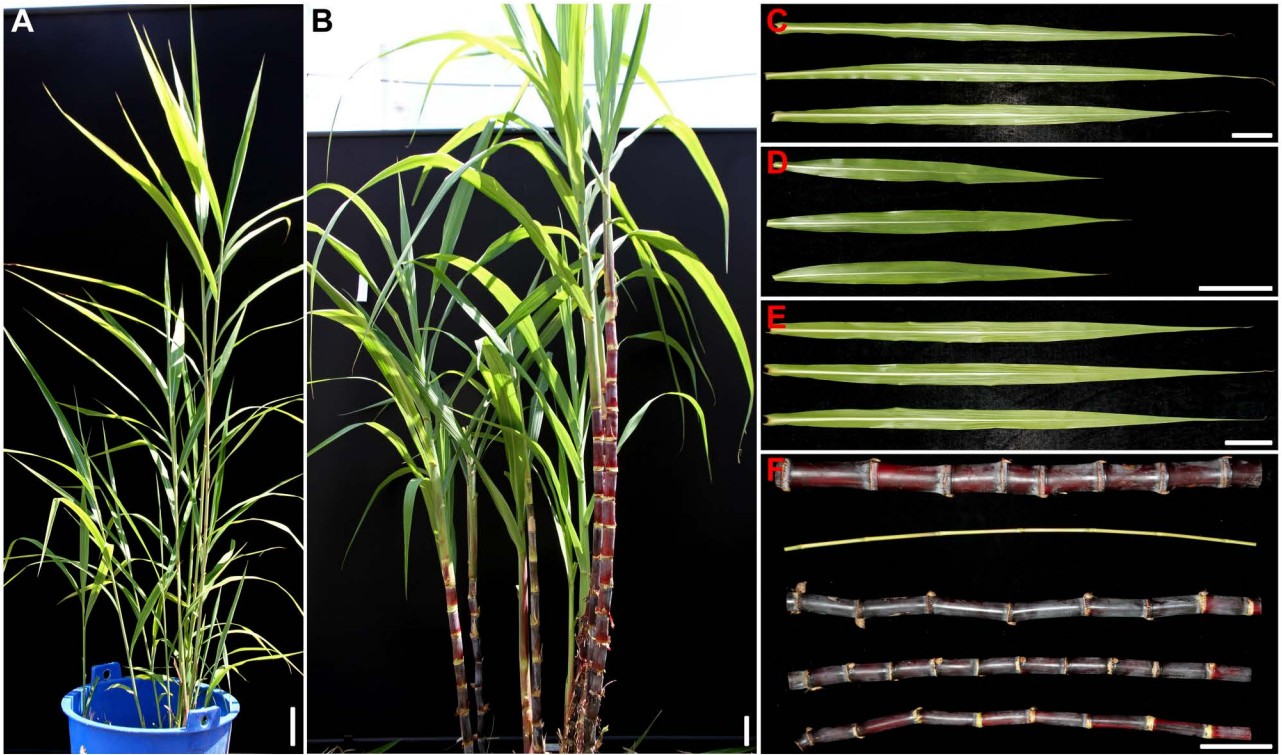

**Fig 8. Plant phenotypes of intergeneric hybrids between *S. officinarum* and *E. rockii*. (A, B)** Plants from *E. rockii* 'Yunnan 15-7' **(A)** and F$_1$ intergeneric hybrids from 'Badila' × 'Yunnan 15-7' **(B)**. **(C-E)** Leaves from 'Badila' **(C)**, 'Yunnan 15-7' **(D)**, and their F$_1$ intergeneric hybrids **(E)**. **(F)** Canes from 'Badila', 'Yunnan 15-7', and their F$_1$ intergeneric hybrids, shown sequentially from top to bottom. Bars = 10 cm.

**Table 2. Main agronomic traits of intergeneric hybrids between *S. officinarum* and *E. rockii*.**

| Material name | Stalk length (cm) | Internode number | Leaf length (cm) | Leaf width (cm) | Stalk diameter (mm) | Juice brix % |
|---|---|---|---|---|---|---|
| Badila | 210.73 ± 13.47 | 20.17 ± 1.03 | 136.27 ± 6.81 | 5.47 ± 0.27 | 39.60 ± 1.40 | 20.79 ± 0.60 |
| Yunnan 15–7 | 125.87 ± 11.40 | 12.92 ± 0.79 | 47.80 ± 3.16 | 3.44 ± 0.21 | 6.01 ± 1.06 | – |
| Badila × Yunnan 15–7 | 163.87 ± 8.11** | 18.47 ± 0.92** | 108.40 ± 7.32** | 5.29 ± 0.27 | 24.73 ± 2.63** | 13.06 ± 0.46** |

Data and errors bars represent mean ± SD. ** Significant difference of F$_1$ hybrids in Student's *t* test (P < 0.01).

diversity [7,34]. In this study, sixteen F₁ hybrids were generated from the intergeneric cross between *S. officinarum* ('Badila', female) and *E. rockii* ('Yunnan 15-7', male) (Fig 8). Traditional methods for identifying *Saccharum* intergeneric hybrids, which rely primarily on morphological characteristics, frequently produce ambiguous results, as distinguishing genuine hybrids from a high-frequency selfed progeny of the female *Saccharum* parent is challenging. In contrast, molecular marker methods, such as RAPD, AFLP, and SSR, have increasingly been employed as conventional techniques for identifying hybrid crosses between *Saccharum* and its related genera over the past two decades [6,7,9,10,34]. However, these markers depend on the presence of multiple amplification bands, which can produce variable results in sugarcane germplasm authentication. Several characteristics of nrDNA-ITS make it a valuable DNA barcode for evaluating evolutionary relationships and distinguishing between *Saccharum* and its related genera, such as high variability, rapid evolutionary rate, and ease of PCR amplification and sequencing [23,24,35]. In this study, a 677 bp fragment was amplified using the general primers ITS1 and ITS4, encompassing the entire ITS sequence (Fig 2). SNPs in the nrDNA-ITS sequence have been identified in numerous crops and have served as valuable molecular markers for identifying interspecies germplasms [36], thereby contributing to molecular breeding strategies. Our results demonstrate that the SNPs at positions 370 and 371 bp in the nrDNA-ITS region are reliable in distinguishing *S. officinarum* from *E. rockii* (Fig 3 and Table 1).

Tetra-primer ARMS-PCR is a PCR-based technique designed specifically for SNP detection [12]. Based on the SNP site in the nrDNA-ITS region, previous studies have successfully applied this method to distinguish *S. spontaneum* and *S. officinarum* [28–30]. In this study, tetra-primer ARMS-PCR revealed that genuine hybrids generated three distinct amplification bands (647 bp, 390 bp, 298 bp), while false hybrids produced two bands (647 bp, 390 bp), consistent with the maternal parent (Fig 5). These findings underscore tetra-primer ARMS-PCR as an efficient and cost-effective tool for identifying intergeneric hybrids of *S. officinarum* and *E. rockii*. HRM curve analysis has become widely adopted in crop genotyping since its development in 2003 [37]. This method analyzes DNA amplicons post-PCR without additional processing and provides results within minutes [17]. In this study, HRM curve analysis with the developed ITS primers effectively differentiated parental homozygotes, heterozygous F₁ intergeneric hybrids, and false hybrids based on distinct melting curve patterns (Fig 6b,c). Our results confirm that tetra-primer ARMS-PCR and HRM curve analysis based on the SNPs at base 370 and 371 of the nrDNA-ITS sequences are reliable for identifying true hybrids of *S. officinarum* and *E. rockii*, despite shared morphological traits with maternal parent (Fig 8). Additionally, GISH has proven effective in analyzing chromosome composition in intergeneric hybrids of *Saccharum* and related genera [38,39]. In this study, GISH revealed that F₁ hybrids confirmed as genuine by molecular markers contained 55 chromosomes, with 40 originating from *S. officinarum* and 15 from *E. rockii*, following an n + n inheritance pattern (Fig 7). Previous research by Lin et al. [40], using GISH characterization, also demonstrated that chromosome transmission follows an n + n pattern in F₁ intergeneric hybrids of *S. officinarum* × *E. rockii*. These findings align with tetra-primer ARMS-PCR and HRM results, further validating the reliability of those molecular techniques for hybrid identification.

Although previous studies have reported that *E. rockii* possesses drought and cold tolerance and good ratooning ability [6], and that its hybrids with *Saccharum* exhibit resistance to brown rust disease [8], the F₁ hybrids from its cross with *S. officinarum* in this study were inferior to the maternal parent in traits such as stalk length, stalk diameter, and sugar content. Therefore, our subsequent breeding program requires a backcrossing strategy with *S. officinarum* or other hybrids to develop new varieties with improved adaptability and commercially acceptable yields and sucrose levels.

## Conclusions

We identified true intergeneric hybrids of *S. officinarum* and *E. rockii* using tetra-primer ARMS-PCR and HRM curve analysis based on SNPs at positions 370 and 371 bp in the nrDNA-ITS sequence. Our findings demonstrate that nrDNA-ITS barcoding, combined with tetra-primer ARMS-PCR and HRM, constitutes an efficient and reliable approach for hybrid identification. GISH assays provided further validation by confirming the chromosome composition of the hybrids. This study lays the groundwork for integrating *E. rockii* traits into modern sugarcane cultivars and underscores the

effectiveness of nrDNA-ITS barcoding combined with tetra-primer ARMS-PCR and HRM analysis in molecular-assisted hybrid identification.

## Supporting information

**S1 File. Source Data Table 2.**
(XLSX)

## Acknowledgments

The authors express their sincere gratitude to Professor Zuhu Deng from Fujian Agriculture and Forestry University for his invaluable assistance in conducting the GISH assay.

## Author contributions

**Conceptualization:** Gang Wang, Hailong Chang.

**Data curation:** Qinggan Liang, Yuanxia Qin, Qingdan Wu, Jiantao Wu.

**Funding acquisition:** Qinnan Wang.

**Investigation:** Gang Wang, Wei Zhang, Huanying Xu, Yuanxia Qin, Qingdan Wu, Cheng Fu, Feng Zhou.

**Methodology:** Jiantao Wu, Cheng Fu, Yuxing An.

**Project administration:** Qinnan Wang, Hailong Chang.

**Resources:** Wei Zhang, Huanying Xu.

**Supervision:** Yuxing An.

**Validation:** Feng Zhou.

**Visualization:** Jianqiang Wang.

**Writing – original draft:** Gang Wang.

**Writing – review & editing:** Jianqiang Wang, Qinggan Liang.

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
