## [Decision Letter · Decision Letter 0]

2 Mar 2025

PONE-D-24-60083Identification and validation of intergeneric hybrids between Saccharum officinarum and Erianthus rockii using molecular and cytogenetic toolsPLOS ONE

Dear Dr. Chang,

Thank you for submitting your manuscript to PLOS ONE. After careful consideration, we feel that it has merit but does not fully meet PLOS ONE’s publication criteria as it currently stands. Therefore, we invite you to submit a revised version of the manuscript that addresses the points raised during the review process.

The editor ask the authors to respond only to the comments raised by Reviewer 1.

We look forward to receiving your revised manuscript.

Kind regards,

Rongchun Han

Academic Editor

PLOS ONE

The authors express their sincere gratitude to Professor Zuhu Deng from Fujian Agriculture and Forestry University for his invaluable assistance in conducting the GISH assay. This research was funded by GDAS’ Project of Science and Technology Development (2022GDASZH-2022010102), the GDAS’s Project of Technical Innovation and Incubation Service Platform Construction (2021GDASYL 20210301001) and was supported by Provincial scientific research institutions stability support sub-project in 2020 "Breeding and construction of healthy seedling propagation system of new sugarcane varieties (lines)".

Reviewers' comments:

Reviewer's Responses to Questions

**Comments to the Author**

1. Is the manuscript technically sound, and do the data support the conclusions?

Reviewer #1: Yes

Reviewer #2: Yes

2. Has the statistical analysis been performed appropriately and rigorously? 

Reviewer #1: Yes

Reviewer #2: Yes

3. Have the authors made all data underlying the findings in their manuscript fully available?

Reviewer #1: Yes

Reviewer #2: Yes

4. Is the manuscript presented in an intelligible fashion and written in standard English?

Reviewer #1: Yes

Reviewer #2: Yes

5. Review Comments to the Author

Reviewer #1: Wang and colleagues proposed a combination of molecular techniques, namely tetra-primer Amplification Refractory Mutation System (ARMS)-PCR, High-Resolution Melting (HRM) curve analysis and Genomic In situ Hybridization (GISH) techniques, in validating true hybrids between Saccharum officinarum and Erianthus rockii. SNPs in the full-length ITS were identified and screened for ARMS and HRM verification. The methods are logical, and the results are valid. The authors paid efforts in improving the experimental design to raise the efficiency and convenience in hybrid identification using each molecular tool. Meanwhile, they include the data from traditional method as morphological identification in this study, making the ground of using molecular tools more sounded. The research outcomes will largely contribute to the breeding programme of sugarcanes.

I have only a one major comment regarding to the study design. The three methods applied in this study have been used in other studies separately as cited (Cai et al. 2005, Aitken et al. 2007, Nair et al. 1999, Devarumath et al. 2012; Lu et al. 2012 and Lin et al. 2013). Although the authors optimized ARMS and HRM for simplified setup and shortened analysis time, I did not see the solid needs to apply all three methods as an integral. To me, the electrophoretogram of figure 5 is already evident for assessing the validity of hybrids. Could the authors explain and elaborate more why the three methods should be applied collectively? Is there any limitation of each method? Also, the GISH technique reveals the composition and the structure of chromosomes and in this case perfectly showed the n+n compositions of true hybrids. However, aneuploid is common in plants which are hybridized or mutated. Did the author considered the aneuploid issues when using this cytogenetic tool?

In addition, a few minor comments the authors should settle.

1) The authority of Saccharum officinarum should be Linnaeus. Is that "Badila" is a cultivar or a line? If so, the cultivar name should be enclosed in a pair of single quotation marks i.e. 'Badila', following the International Code of Nomenclature for Cultivated Plants (ICNCP). Also, please do so for "Yunnan 15-7" if this situation is applied.

2) In line 281, the titles of Fig 6b ("the normalized fluorescence curves") and Fig 6c ("the difference plot melt curve") are not consistent with the one in Figure 6 ("Normalized Melting Curves" and "Normalized and Temp-Shifted Difference Curves"). Please be consistent for easy comprehension.

3) In Table 1, the primer names (R-D2 5'-3' RI-Dian4 5'-3') are not identical with those (FI-Er, RI-So) in the main text. Please check and revise.

Reviewer #2: The study provides significant insights into sugarcane breeding by utilizing molecular and cytogenetic tools to confirm intergeneric hybrids between Saccharum officinarum and Erianthus rockii. The integration of tetra-primer ARMS-PCR, HRM curve analysis, and GISH ensures precise identification of true hybrids, addressing a critical challenge in distant hybridization. The findings highlight a distinct chromosome inheritance pattern and demonstrate the potential of E. rockii for enhancing genetic diversity in sugarcane. However, the reduced yield and sugar content in hybrids suggest the need for further backcrossing to improve agronomic traits. The study effectively establishes a foundation for utilizing E. rockii traits in modern sugarcane breeding and underscores the efficiency of nrDNA-ITS barcoding and molecular-assisted hybrid identification techniques.

6. PLOS authors have the option to publish the peer review history of their article (what does this mean? ). If published, this will include your full peer review and any attached files.

**Do you want your identity to be public for this peer review?** For information about this choice, including consent withdrawal, please see our Privacy Policy .

Reviewer #1: No

Reviewer #2: No

---

## [Author Response · Author response to Decision Letter 1]

4 Apr 2025

Response to Reviewer 1 Comments

Major comment: I have only a one major comment regarding to the study design. The three methods applied in this study have been used in other studies separately as cited (Cai et al. 2005, Aitken et al. 2007, Nair et al. 1999, Devarumath et al. 2012; Lu et al. 2012 and Lin et al. 2013). Although the authors optimized ARMS and HRM for simplified setup and shortened analysis time, I did not see the solid needs to apply all three methods as an integral. To me, the electrophoretogram of figure 5 is already evident for assessing the validity of hybrids. Could the authors explain and elaborate more why the three methods should be applied collectively? Is there any limitation of each method? Also, the GISH technique reveals the composition and the structure of chromosomes and in this case perfectly showed the n+n compositions of true hybrids. However, aneuploid is common in plants which are hybridized or mutated. Did the author considered the aneuploid issues when using this cytogenetic tool?

Response: Thank you for your comments. In response to your three queries, I will address them individually.

1. In general, I agree with your observation that the authenticity of hybrids can be verified solely through ARMS-PCR using the developed molecular marker. However, for any newly developed molecular marker, its reliability should be validated through additional methods. Once the reliability of a newly developed molecular marker has been confirmed through multiple methods, it becomes redundant to apply all three methods simultaneously in every case. Only ARMS-PCR, based on the developed molecular markers, is required for hybrid authenticity verification.

2. All three techniques have limitations. ARMS-PCR requires high specificity for primers and amplification conditions, necessitating numerous pre-experiments to identify optimal conditions. HRM equipment and reagents are relatively costly. The GISH experimental process is complex and not suitable for large-scale identification.

3. S. officinarum is autopolyploid with 2n = 8x = 80, while E. rockii is diploid with 2n = 30. The number of gametic chromosomes produced during meiosis is typically euploidy halved, so the chromosome number of F1 hybrids typically conforms to the n + n genetic pattern. However, when F1 hybrids are crossed further, they may produce gametes with aneuploidy halving the number of chromosomes, and the chromosome numbers of their offspring will exhibit deviations from the n + n pattern.

Minor comments 1: The authority of Saccharum officinarum should be Linnaeus. Is that "Badila" is a cultivar or a line? If so, the cultivar name should be enclosed in a pair of single quotation marks i.e. 'Badila', following the International Code of Nomenclature for Cultivated Plants (ICNCP). Also, please do so for "Yunnan 15-7" if this situation is applied.

Response: Agree. “Badila” is a cultivar, and “Yunnan 15-7” is a wild material found in the fields of Yunnan, China. We have, accordingly, enclosed all “Badila” and “Yunnan 15-7” in the manuscript with a pair of single quotation marks. The changes can be found in the revised manuscript lines 111-119, 158, 200-201, 225, 229, 239, 269, 298, 307-317 and 335.

Minor comments 2: In line 281, the titles of Fig 6b ("the normalized fluorescence curves") and Fig 6c ("the difference plot melt curve") are not consistent with the one in Figure 6 ("Normalized Melting Curves" and "Normalized and Temp-Shifted Difference Curves"). Please be consistent for easy comprehension.

Response: Agree. We have, accordingly, modified the content in the HRM assay identification of true and false F1 hybrids of the Results to emphasize this point. Line 281 to 282 in the revised manuscript this change can be found.

Minor comments 3: In Table 1, the primer names (R-D2 5'-3' RI-Dian4 5'-3') are not identical with those (FI-Er, RI-So) in the main text. Please check and revise.

Response: Agree. We have, accordingly, modified the primer names in Table 1 to emphasize this point. Table 1 in the revised manuscript this change can be found.

Response to Academic Editor Comments

Comment 1: Please ensure that your manuscript meets PLOS ONE's style requirements, including those for file naming. The PLOS ONE style templates can be found at https://journals.plos.org/plosone/s/file?id=wjVg/PLOSOne_formatting_sample_main_body.pdf and https://journals.plos.org/plosone/s/file?id=ba62/PLOSOne_formatting_sample_title_authors_affiliations.pdf

Response: Agree. We have, accordingly, changed the font size of headings at all levels to emphasize this point. In the revised manuscript this change can be found.

Comment 2: We note that the grant information you provided in the ‘Funding Information’ and ‘Financial Disclosure’ sections do not match. When you resubmit, please ensure that you provide the correct grant numbers for the awards you received for your study in the ‘Funding Information’ section.

Response: Agree. We have, accordingly, added the following funding statement content in the ‘Funding Information’ section to emphasize this point. The following text is the funding statement: This research was funded by GDAS’ Project of Science and Technology Development (2022GDASZH-2022010102), the GDAS’s Project of Technical Innovation and Incubation Service Platform Construction (2021GDASYL 20210301001). The funders had no role in study design, data collection and analysis, decision to publish, or preparation of the manuscript.

Comment 3: Thank you for stating the following in the Acknowledgments Section of your manuscript:

The authors express their sincere gratitude to Professor Zuhu Deng from Fujian Agriculture and Forestry University for his invaluable assistance in conducting the GISH assay. This research was funded by GDAS’ Project of Science and Technology Development (2022GDASZH-2022010102), the GDAS’s Project of Technical Innovation and Incubation Service Platform Construction (2021GDASYL 20210301001) and was supported by Provincial scientific research institutions stability support sub-project in 2020 "Breeding and construction of healthy seedling propagation system of new sugarcane varieties (lines)".

Response: Agree. We have, accordingly, removed funding-related text in the Acknowledgments section to emphasize this point. Line 400 to 402 in the revised manuscript this change can be found.

Comment 4: We note that your Data Availability Statement is currently as follows: All relevant data are within the manuscript and its Supporting Information files.

Response: Agree. We have, accordingly, presented raw data of Table 2 as a supporting information S1 File to emphasize this point. Line 613 to 614 in the revised manuscript this change can be found.

---

## [Decision Letter · Decision Letter 1]

22 Apr 2025

Identification and validation of intergeneric hybrids between Saccharum officinarum and Erianthus rockii using molecular and cytogenetic tools

PONE-D-24-60083R1

Dear Dr. Chang,

We’re pleased to inform you that your manuscript has been judged scientifically suitable for publication and will be formally accepted for publication once it meets all outstanding technical requirements.

Kind regards,

Rongchun Han

Academic Editor

PLOS ONE

Additional Editor Comments (optional):

Reviewers' comments:

Reviewer's Responses to Questions

**Comments to the Author**

1. If the authors have adequately addressed your comments raised in a previous round of review and you feel that this manuscript is now acceptable for publication, you may indicate that here to bypass the “Comments to the Author” section, enter your conflict of interest statement in the “Confidential to Editor” section, and submit your "Accept" recommendation.

Reviewer #1: All comments have been addressed

2. Is the manuscript technically sound, and do the data support the conclusions?

Reviewer #1: Yes

3. Has the statistical analysis been performed appropriately and rigorously? 

Reviewer #1: Yes

4. Have the authors made all data underlying the findings in their manuscript fully available?

Reviewer #1: Yes

5. Is the manuscript presented in an intelligible fashion and written in standard English?

Reviewer #1: Yes

6. Review Comments to the Author

Reviewer #1: The authors have addressed my comments adequately. Although I still feel that it is a bit overdoing by combining all three approaches together. I feel the work is technically sound and hence I recommend its publication.

7. PLOS authors have the option to publish the peer review history of their article (what does this mean? ). If published, this will include your full peer review and any attached files.

**Do you want your identity to be public for this peer review?** For information about this choice, including consent withdrawal, please see our Privacy Policy .

Reviewer #1: No

---

## [Editor Report · Acceptance letter]

PONE-D-24-60083R1

PLOS ONE

Dear Dr. Chang,

I'm pleased to inform you that your manuscript has been deemed suitable for publication in PLOS ONE. Congratulations! Your manuscript is now being handed over to our production team.

Kind regards,

on behalf of

Prof Dr Rongchun Han

Academic Editor

PLOS ONE